# Deciphering the Longevity and Levels of SARS-CoV-2 Antibodies in Children: A Year-Long Study Highlighting Clinical Phenotypes and Age-Related Variations

**DOI:** 10.3390/pathogens13080622

**Published:** 2024-07-26

**Authors:** Gemma Pons-Tomàs, Rosa Pino, Aleix Soler-García, Cristian Launes, Irene Martínez-de-Albeniz, María Ríos-Barnés, Maria Melé-Casas, María Hernández-García, Manuel Monsonís, Amadeu Gené, Mariona-F. de-Sevilla, Juan-José García-García, Claudia Fortuny, Victoria Fumadó

**Affiliations:** 1Paediatric Department, Hospital Sant Joan de Déu, University of Barcelona, 08950 Barcelona, Spain; gemma.pons@sjd.es (G.P.-T.); rosamaria.pino@sjd.es (R.P.); aleix.soler@sjd.es (A.S.-G.); maria.mele@sjd.es (M.M.-C.); maria.hernandezg@sjd.es (M.H.-G.); mariona.fernandez@sjd.es (M.-F.d.-S.); juanjose.garciag@sjd.es (J.-J.G.-G.); 2Infectious Diseases and Microbiome Research Group, Institut de Recerca Sant Joan de Déu (IRSJD), 08950 Barcelona, Spain; maria.rios@sjd.es (M.R.-B.); claudia.fortuny@sjd.es (C.F.); victoria.fumado@sjd.es (V.F.); 3Department of Surgery and Medical-Surgical Specialties, Faculty of Medicine and Health Sciences, University of Barcelona, 08036 Barcelona, Spain; 4Consorcio de Investigación Biomédica en Red de Epidemiología y Salud Pública (CIBERESP), 28029 Madrid, Spain; 5Infectious and Imported Diseases Department, Hospital Sant Joan de Déu, 08950 Barcelona, Spain; irene.malbeniz@gmail.com; 6Department of Microbiology, Hospital Sant Joan de Déu, 08950 Barcelona, Spain; manuel.monsonis@sjd.es (M.M.); amadeu.gene@sjd.es (A.G.)

**Keywords:** SARS-CoV-2, serology, antibodies, paediatrics, hospitalisation

## Abstract

Background: Identifying potential factors correlated with the sustained presence of antibodies in plasma may facilitate improved retrospective diagnoses and aid in the appraisal of pertinent vaccination strategies for various demographic groups. The main objective was to describe the persistence of anti-spike IgG one year after diagnosis in children and analyse its levels in relation to epidemiological and clinical variables. Methods: A prospective, longitudinal, observational study was conducted in a university reference hospital in the Metropolitan Region of Barcelona (Spain) (March 2020–May 2021). This study included patients under 18 years of age with SARS-CoV-2 infection (positive PCR or antigen tests for SARS-CoV-2). Clinical and serological follow-up one year after infection was performed. Results: We included 102 patients with a median age of 8.8 years. Anti-spike IgG was positive in 98/102 (96%) 12 months after the infection. There were higher anti-spike IgG levels were noted in patients younger than 2 years (*p* = 0.034) and those with pneumonia (*p* < 0.001). A positive and significant correlation was observed between C-reactive protein at diagnosis and anti-spike IgG titre one-year after diagnosis (*p* = 0.027). Conclusion: Anti-SARS-CoV-2 IgG antibodies were detected in almost all paediatric patients one year after infection. We also observed a positive correlation between virus-specific IgG antibody titres with SARS-CoV-2 clinical phenotype (pneumonia) and age (under 2 years old).

## 1. Introduction

COVID-19, caused by the SARS-CoV-2 virus, posed a global public health challenge due to its rapid spread and impact on society. From the beginning of the pandemic, a large number of scientific articles described its more severe involvement in adults compared to the paediatric population [1]. However, in April 2020, the first case of multisystem inflammatory syndrome (MIS-C) related to the new virus was documented in a paediatric patient, which, although rare, is a severe clinical presentation [2].

Considering the health and social crisis caused by SARS-CoV-2 infection, the serological response after infection is one of the key points of research. Multiple studies have described the importance of humoral immunity within adaptive immunity in the control of SARS-CoV-2 infection. Four main structural proteins involved in infection have been identified [3,4] (spike (S), envelope (E), membrane (M) and nucleocapsid (N)). Along with the M protein, the S protein is highly immunogenic and is the main target of vaccines [4]. The N and S proteins are highly conserved structural proteins in the *Coronaviridae* family [4,5,6], although some studies have observed greater cross-reactivity with the N protein [7]. The S protein has two domains: S1 and S2. The S1 domain includes the receptor-binding domain (RBD) [4]. Through the S protein, the virus enters the host cell via the ACE2 [4,8] receptor.

Serum antibodies IgA, IgM, and IgG, and in particular anti-RBD, are involved in the immune response against SARS-CoV-2. Anti-RBD IgG and anti-spike IgG are reliable predictors of virus neutralization, a role also shared by plasma and mucosal IgA and IgM [9,10].

Understanding the protection after infection and longevity of antibodies is crucial to control the spread of infection and devise population vaccination strategies. This involves determining when the serological response begins, the duration of antibodies in the blood and identifying any factors that might influence it.

It has been reported that a large proportion of people with a history of SARS-CoV-2 infection develop IgG antibodies between the first and third week after infection [10,11], including those who are asymptomatic. Anti-spike IgG antibodies have been observed to persist in patients’ blood for several months following the infection, providing some level of protection against the virus [12,13]. Anti-nucleocapsid antibodies, however, are less enduring [14]. Data on the longevity of the humoral response against SARS-CoV-2 in the paediatric population remain limited. Moreover, studying antibody persistence in children could be more informative than in adults as children are less likely to have cross-reactive antibodies from previous coronavirus infections [15].

Vaccination in adults started in December 2020 in Spain. In the paediatric population, it started a year later, and its implementation was more controversial due to the benign course of this disease in most children. Given that humoral immunity in children differs from adults, specific studies are required to determine the duration of immunity and identify potential factors associated with the sustained presence of antibodies in plasma. This would facilitate improved retrospective diagnoses and aid in the appraisal of pertinent vaccination strategies for various demographic groups.

### Objectives

The main objective of this study was to describe the persistence of anti-spike IgG antibodies one year after SARS-CoV-2 infection in children and analyse their levels in relation to epidemiological and clinical variables. A description of the levels of antibodies within 1 month and 6 months of infection would be also described, if available.

## 2. Materials and Methods

### 2.1. Description of the Study

A prospective, longitudinal, observational study was conducted in a university reference hospital located in the Metropolitan Region of Barcelona (Spain). This study included patients under 18 years of age with SARS-CoV-2 infection, confirmed by positive PCR or antigen tests for SARS-CoV-2. Patients were recruited through the following methods:Patients visiting the hospital with fever or respiratory symptoms.Asymptomatic patients attending pre-surgical screenings, admitted for non-SARS-CoV-2-related symptoms, or screened due to contact with a symptomatic case.Admitted patients with fever and/or respiratory symptoms.Admitted patients showing symptoms compatible with MIS-C.

Some patients were included due to routine clinical protocols, while others were part of the Kids Corona platform, a platform specifically created in order to investigate and understand the impact of SARS-CoV-2 on children [16,17,18,19,20]. The study included those who agreed to clinical and serological follow-up one year after infection. Participants whose parents did not provide consent were excluded.

The inclusion period spanned from March 2020 to May 2021. This timeframe occurred before the initiation of vaccinations in children, during which the Wuhan variant was predominant.

### 2.2. Variables

Clinical and serological variables were collected and analysed. Among the clinical variables, data were collected on age, sex, personal and family history, symptoms at diagnosis (present or absent and type of symptoms), the need for admission (including the need for admission in the Intensive Care Unit (ICU)), reason for admission and information on follow-up (data on reinfection and persistent symptoms 12 weeks after infection). The primary result was the measurement of venous anti-spike IgG levels one year after diagnosis. However, some patients were also evaluated for capillary or venous levels of anti-nucleocapsid IgM, anti-nucleocapsid IgG, or anti-spike at intermediate points, such as within the first month or three months after diagnosis.

A follow-up visit was conducted one year after infection to inquire about the patient’s clinical progress. Information was obtained from both the study centre and the primary care centre the patient visited.

### 2.3. Laboratory

For the main outcome (serologic response one year after initial diagnosis), serum samples obtained by venopuncture were tested using a chemiluminescent microparticle immunoassay (CMIA). Specifically, the SARS-CoV-2 IgG II Quant assay (Abbott) on the ARCHITECT i Systems was employed. This assay is designed to detect and quantify immunoglobulins of class G (IgG), including neutralising antibodies, against the receptor-binding domain (RBD) region of the S1 subunit of the SARS-CoV-2 spike protein in serum and plasma. Antibody quantification is expressed in binding units per millilitre (AU/mL), with a negative result considered at <50 AU/mL. Only one replicate per sample was run, and the positivity thresholds were determined according to the assay standards specified by the manufacturer.

For intermediate points, different tests were utilised due to the limitations of techniques available during the initial pandemic phase. Capillary results were obtained using a rapid test based on lateral flow immunoassay (Sure Screen COVID-19 IgG/IgM), conducted according to the manufacturer’s instructions on capillary blood samples taken by finger prick. When venopuncture results were reported, they were based on the CMIA assay Abbott SARS-CoV-2 IgG assay on the ARCHITECT i Systems, which provided qualitative results for specific IgM against spike glycoprotein and IgG against nucleocapsid. IgM antibodies against the spike glycoprotein were considered positive if they had an index value of 1.0 or higher. For IgG nucleocapsid antibodies, a threshold of 1.4 was used to distinguish between negative and positive results. From January 2021, quantitative IgG against spike glycoprotein (instead of anti-nucleocapsid) was also determined using the SARS-CoV-2 IgG II Quant assay. Again, only one replicate per sample was performed, and the positivity thresholds were determined based on each product’s technical specifications and assay standards.

### 2.4. Statistics

Categorical variables are represented as total numbers and proportions, whereas continuous variables are described using medians and interquartile ranges (IQRs). The Mann–Whitney U test and Kruskal–Wallis test were used to identify differences between groups in quantitative variables. Spearman’s Rho test was utilised to determine the correlation between continuous quantitative variables. Confidence intervals (CIs) were set at 95%. Statistical significance was determined at *p* < 0.05. These statistical analyses were carried out using IBM SPSS Statistics for Windows, version 27.0.1.

A receiver operator characteristic (ROC) curve analysis was conducted using R software (version 4.3.2; R Foundation, Vienna, Austria) and the pROC library v1.18.5. This analysis differentiated between patients who had pneumonia and those who were asymptomatic or had other clinical syndromes based on anti-spike IgG titres. The Youden point marked the value of antibodies with the highest average specificity and sensitivity.

### 2.5. Ethics

The study was approved by the Ethics Committee of the Institut de Recerca Sant Joan de Déu (EOM-90-21). Signed informed consent was obtained from all participants, from one or both parents or their legal guardians.

## 3. Results

### 3.1. Study Population

During the study period, 102 patients with a median age of 8.8 years (IQR 3.8–12.7) were included, of whom 57/102 (56%) were male. A total of 2/102 (2%) had a relevant comorbidity that could act as a risk factor for a more severe respiratory infection (common variable immunodeficiency and bronchiectasis in the context of genetic syndrome). Among the included patients, 46/102 (45%) were asymptomatic (either as contacts of a confirmed case or through presurgical screening), and 43/102 (42%) exhibited fever and/or respiratory symptoms (16 of them were admitted, of whom 9 had lower respiratory tract infection). Moreover, 13/102 (13%) presented with symptoms compatible with MIS-C associated with SARS-CoV-2 infection. The most frequent symptoms at diagnosis were fever (50/102; 49%), cough (31/102; 30%) and asthenia (22, 22%). One patient required admission to the ICU due to severe pneumonia, requiring respiratory support with non-invasive mechanical ventilation and inotropic drugs.

### 3.2. Antibody Levels One Year after Confirmed Infection and Association with Epidemiological and Clinical Variables

The presence of antibodies in venopuncture specimens was obtained at a median of 12 months (IQR: 11–13). Among the entire group of sampled patients, anti-spike IgG was positive in 98/102 (96%), with median levels of 671 AU/mL (IQR: 376–1443).

The relationship between anti-spike IgG levels and the variables of sex, age, clinical phenotype and analytical parameters was studied (Table 1). Regarding age, higher anti-spike IgG levels in patients younger than 2 years (*p* = 0.034) were observed when compared to the rest of the age groups. After multivariate analysis, a trend towards significance was also observed (*p* = 0.054).

Patients were classified according to clinical phenotype at diagnosis (asymptomatic, fever +/− upper respiratory tract infection, lower respiratory tract infection (pneumonia) and MIS-C) and higher levels were observed in patients with pneumonia in the multivariate analysis, with a median level of 3012 AU/mL (IQR: 1406–6671) (*p* < 0.001). Anti-spike IgG levels one year after infection were accurate in distinguishing patients who had had pneumonia a year prior from those who did not (area under the ROC curve: 0.796; *p* < 0.01). A level above 2401 AU/mL was associated with having had pneumonia, with a specificity of 94.6% and a sensitivity of 77.8%. See Figure 1.

Finally, the association between antibody level and analytical parameters at diagnosis could be analysed in 25 patients for whom blood tests results were available. A positive and significant correlation was observed between C-reactive protein at diagnosis and anti-spike IgG titre one year after diagnosis (*p* = 0.027). In contrast, no association was found between antibody level and leukocyte or lymphocyte count. No relation between antibody level and sex was seen.

### 3.3. Clinical Outcomes

None of the patients included were diagnosed with COVID-19 during the follow-up. In five patients, symptoms persisted beyond 12 weeks after diagnosis, with three patients presenting with anosmia and ageusia, one patient with exertional dyspnoea and one patient with abdominal pain and an anxiety disorder. The median anti-spike IgG levels were 670 AU/mL (IQR: 373–1448) in patients with persistent symptoms compared to 793 AU/mL (IQR: 295–2719) in those without them.

### 3.4. Other Serological Results Obtained during the First Year

In 83/102 (81%) patients, a serologic test was performed within the first month after diagnosis (median 6 days, IQR 0–29.5). Of these, capillary tests were conducted on 49 patients (59%), with 15 (31%) testing positive for IgG and 35 (71%) for IgM. Venopuncture was performed on 34 (41%). Among the venous serologies, anti-spike IgG was determined in four patients, all of whom tested positive with values of 17,915,674,796 and 6166 AU/mL. Qualitative anti-nucleocapsid IgG was determined in the remaining 30, with 26 testing positive (87%). Qualitative IgM anti-spike was detected in 7 out of the 14 tested (50%).

Beyond the first month of infection, serology was performed in 30/102 (25%) patients (median 2.7 months after diagnosis, IQR: 1.6–4.7). Capillary tests were conducted on only one patient, with both IgG and IgM testing negative. Venopuncture was performed on 29 (97%). In 11 of them, anti-spike IgG was determined, wherein all tested positive, with a median value of 701 AU/mL (IQR: 586–1960). Qualitative anti-nucleocapsid IgG was determined in the remaining 18, being positive in 16 (89%). Qualitative IgM anti-spike was detected in 3 out of 10 patients (30%).

The four patients who did not meet the positive threshold for anti-spike IgG levels at the primary endpoint (12 months after diagnosis) showed no serologic detection at intermediate points. All of them were previously healthy children, asymptomatic at diagnosis, and were between 1.7 and 9.6 years old.

## 4. Discussion

Infections from common cold coronaviruses (HCoV-OC43 and HCoV-HKU1), as well as SARS-CoV-1, induce immunity against reinfection for varying periods. For instance, some studies have shown that immunity may last for at least a year, while other observations suggest that immunity can decline more rapidly depending on individual differences and the specific virus involved [21]. It is important to distinguish between the presence of antibodies and the prevention of disease, as antibodies may persist even if reinfections occur [21,22]. Studies on adults have shown that serologic response following SARS-CoV-2 infection persists for at least 12 months [23]. But data on the longevity of antibodies against SARS-CoV-2 in the paediatric population remain limited. This study, one of the largest of its kind in the international literature, shows that almost all children who had experienced SARS-CoV-2 infection during the first wave of the pandemic were found to still have antibodies after 12 months. Notably, those who had lower respiratory tract infections had higher antibody levels.

Long-term persistence has also been seen in smaller studies involving children. In the Dunay et al. cohort [24], 67 children who had a mild infection (outpatient follow-up) were studied along with their household, and positive antibodies (anti-spike IgG and anti-nucleocapsid IgA/IgG/IgM) were detected in the children throughout the study (up to 270 days post infection). Similar results were also obtained in other smaller cohorts observing the persistence of antibodies 10–12 months after infection [25,26]. Most of these studies have focused on patients with asymptomatic or mild infection (outpatient management), with very few including severe paediatric patients. A recent study [27] included paediatric hospitalised patients with different degrees of disease severity. This study analysed the evolution of seropositivity as well as antibody titres during the first year after infection. Antibody persistence after one year of infection was observed in most of the included children. However, it is important to note that, at the end of follow-up (>10 months), antibodies were studied in only 12 patients. Therefore, we believe that one strength of our study is the inclusion of a larger sample of paediatric patients with different degrees of disease severity.

Despite the persistence of antibodies, it is unclear whether their neutralising capacity, and thus the protective capacity, is maintained over time [28]. The meta-analysis conducted by Flacco et al. concluded that the reinfection rate in patients not vaccinated during the first wave was low (0.74%) after 12 months [29]. These data are in line with our study in which no reinfections were detected during follow-up. However, these data should be interpreted with caution given the non-negligible proportion of asymptomatic infections in children.

The relationship between COVID-19 severity and long-term antibody level is poorly understood and the mechanisms underlying this association remain unclear. Reports suggest that patients with severe symptoms often have higher viral loads [30], and other research mainly involving adult patients also indicates a correlation between symptom severity and stronger serological responses [10,27,31,32]. Hence, it is plausible that initial quantities of viral antigens might contribute to stronger serological responses. Our study revealed a link between the presence of LRTI at diagnosis and a higher antibody titre one year after infection. Specifically, we observed that an antibody level above 2401 AU/mL after one year was associated with having had SARS-CoV-2 pneumonia, exhibiting high sensitivity and specificity. These data could be useful in cases where patients have co-infections from other microorganisms, and it is unclear which agent is primarily causing clinical deterioration or sequelae. Though the study includes a small number of patients with pneumonia, this preliminary evidence could stimulate further research. This could enhance our understanding of the SARS-CoV-2 pneumonia burden in the context of seasonal circulation. The correlation of higher C-reactive protein levels at the time of diagnosis with elevated anti-spike IgG blood levels one year post diagnosis may also reflect a relationship between the severity of the infection and the strength of the serologic response. This specific correlation might suggest that systemic infection is required to stimulate a more robust serologic response [32].

Concerning patients with MIS-C, the fact that immunoglobulins are part of their treatment may imply that the antibodies detected during follow-up could have been transferred through infusion [33]. In any case, we found lower IgG levels in patients with MIS-C compared to patients with pneumonia; so, the history of immunoglobulin transfusion did not appear to be a confounding factor. Lapp et al. [34] also observed a lower antibody titre in the convalescent phase in patients with MIS-C compared to those with acute SARS-CoV-2 infection (contrary to what was observed during hospitalisation, when higher values were observed in patients with MIS-C). One of the factors that could influence a lower antibody level in MIS-C patients at long-term follow-up could be the fact that they receive treatment with immunomodulators such as corticosteroids, and this could influence the long-term immunological response. Genetic attributes may also contribute to the different potency and durability of humoral responses [35].

Since the beginning of the pandemic, the scientific community has been interested in why children have significantly milder symptomatology compared to adults. It has been postulated that one of the factors that could influence this is age-related differences in immune response [36]. Consistent with this, in our study, we also found a trend toward a higher antibody titre in children under 2 years of age after one year of infection compared with adolescents. Human breast milk contains immunoglobulins (IgA, IgG, and IgM) that may prevent local infections, but the impact on infants’ immunoglobulin levels in blood is uncertain [37]. Our results align with prior studies that describe higher antibody levels in young children compared to adolescents and adults [26,38,39].

A limitation of this study is that some patients may have experienced a non-diagnosed mild or asymptomatic reinfection. Our current assay setup does not support the identification of variations in epitope targeting. This means that the antibodies produced early in the children could target different epitopes upon repeated exposures. The use of assays that can identify these variations would give a more complete understanding of the immune response against the virus. On the other hand, we conducted the study with patients who were infected during the first wave (the Wuhan variant). Even though it seems that children develop antibodies that protect them against the same variant, some studies have reported a decrease in immunoreactivity and neutralising potency against new emerging variants. This could potentially increase the rate of reinfection [28,29,40,41]. More research on this issue is needed to elucidate these questions. Finally, our study was limited to investigating anti-spike IgG only. The absence of neutralising antibody tests, anti-spike IgA antibodies, and the determination of the cellular immune response is a significant limitation in evaluating the full immune response against the infection of the COVID-19 disease.

## 5. Conclusions

Almost all paediatric patients were found to have anti-spike IgG antibodies one year after infection. Those with lower respiratory tract infection (a subrogate indicating greater severity) and those under 2 years old had higher antibody titres. Children infected during the first wave appear to have obtained protection for at least 12 months. However, it is unclear if this protection extends to other variants. These findings should be considered when designing vaccination strategies for children.

## Figures and Tables

**Figure 1 pathogens-13-00622-f001:**
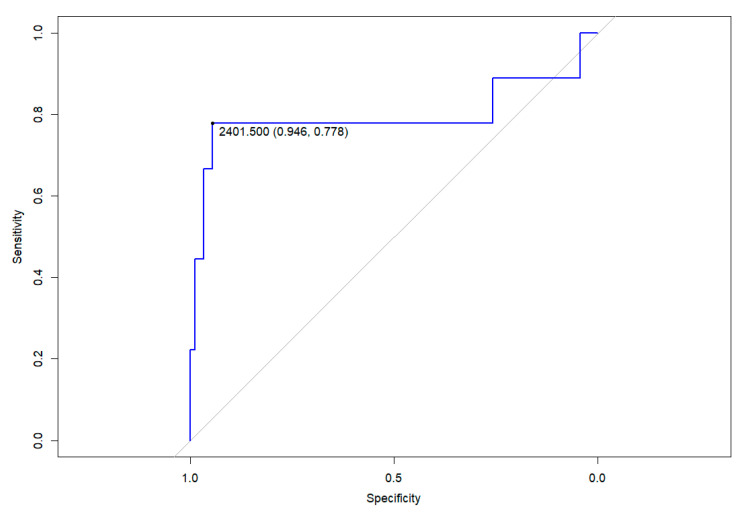
Receiver operating curve for anti-spike IgG levels one year post-diagnosis in discriminating the diagnosis of pneumonia. Mark on the Youden point: value of anti-spike IgG titres in AU/mL (specificity, sensitivity).

**Table 1 pathogens-13-00622-t001:** Associations and correlations between antibody levels one year after confirmed infection and epidemiological and clinical variables.

			Univariate	Multivariable ^ⴕ^	
	**n**	**IgG levels** (AU/mL) 671 (IQR: 376-1443)	** *p* ** **-value**	**Beta coefficient** (95% confidence interval)	** *p* ** **-value**
**Age** <2 years 2–4 5–11 >11	12 22 35 32	1569 (939–2109) 633 (381–1045) 616 (312–997) 649 (312–2176)	0.034 *	−738 (−1487–12) n.a. n.a. n.a.	0.054 n.a. n.a. n.a.
**Sex** Male Female	57 45	670 (339–1619) 672 (415–1217)	0.832	n.a. n.a.	n.a. n.a.
**Clinical Phenotype** Asymptomatic Fever+−URTI LRTI MIS-C	46 34 9 13	593 (249–1066) 640 (338–1327) 3012 (1406–6671) 601 (128–1548)	0.042 ** 0.851 ** 0.003 ** 0.416 **	−309 (−829–210) n.a. 3028 (2112–3945) n.a.	0.240 n.a. <0.001 n.a.
		**Correlations** ** ^ⴕⴕ^ **			
**Blood parameters at diagnosis** Leucocytes Lymphocytes C-Reactive Protein protein	25	−0.212 −0.187 0.461	0.298 0.371 0.027	n.a. n.a. n.a.	n.a. n.a. n.a.

LRTI: lower respiratory tract infection; MIS-C: multisystemic inflammatory syndrome in children; n.a. not applicable; URTI: upper respiratory tract infection. * Patients under 2 years old vs. all others. ** Patients with this condition vs. all others. ^ⴕ^ R-squared of the model is 0.355 ^ⴕⴕ^ Spearman rho.

## Data Availability

The datasets generated and/or analysed during the current study are not publicly available due to concerns that individual privacy could be compromised, but they are available from the corresponding author upon reasonable request.

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
