# Peer review of "Deciphering the Longevity and Levels of SARS-CoV-2 Antibodies in Children: A Year-Long Study Highlighting Clinical Phenotypes and Age-Related Variations"

_pathogens, 2024, doi:10.3390/pathogens13080622_

Round 1

Reviewer 1 Report

Comments and Suggestions for Authors

Pons-Tomàs et al described a concise study on the persistence of antibodies anti-spike IgG one year after SARS-CoV-2 infection in a large cohort of children. The manuscript is well-written and describes an interesting question in the field, however, a few modifications must be addressed.

Introduction: fifth paragraph, the sentence "However, anti-nucleocapsid...." has to be rewritten.

Discussion: The authors have to discuss deeply and justify why they did not test neutralizing antibodies or anti-Spike IgA antibodies. Also, they should include as limitations of the study the absence of the determination of the cellular immune response.

Reviewer 2 Report

Comments and Suggestions for Authors

Summary:

This study by Pons-Tomás, et al., describes the durability of the anti-spike antibody response in children one year after SARS-CoV-2 infection. They also analyzed the association between antibody levels and epidemiological and clinical factors at diagnosis, such as disease severity. The manuscript is clearly written, results are well presented and the discussion puts their results into context.

This study is one of the larger pediatric cohorts reported with over 100 patients included with one-year follow up and serological testing. Within the cohort, they had reasonably good representation across ages and disease groups. The authors note that due to the timing of the study early in the pandemic, the conclusions might differ with new viral variants and vaccinations. The findings they describe are not novel, and they are consistent with numerous reports in adults and children. However, the focus on a pediatric population, the size of the cohort and disease severity groups, annotation and follow up give this study strength; and it would contribute to the body of knowledge about SARS-CoV-2 humoral responses in children.

General comments:

This study addresses a straight-forward question, to assess the durability of anti-spike antibodies in children after SARS-CoV-2 infection. It is somewhat limited in the scope of the science, but the strength of the study is in the cohort. The authors did a nice job of contextualizing their findings within the existing literature. They note that their study adds to the field due to the size of the pediatric cohort and the inclusion of patients with more severe disease. Indeed, one of their key findings is that patients with more severe disease had higher anti-spike IgG levels after a year.

The cohort and methods are mostly well described. However, more detail on the laboratory testing would be needed. How many replicates were run for each? How were the thresholds for positivity determined? Were these based on the assay standards or empirically determined? One assay is listed for IgG anti-spike antibodies, was the same assay used for the other types of antibodies tested?

The other antibody measures (non-spike, non-IgG) and the timing of the interim measures was confusing. These results are a very minor part of the study because of the incomplete testing, but section 3.4 could be clarified. Even with the small sample size, were there any results that could support other results in the study? Any comparisons that could be made? Could there be a supplemental table or figure to present the data more clearly?

To increase the significance, the authors could describe the types of anti-spike antibodies (anti-RBD, anti-S1 or S2 subunit) that persist after a year? This breakdown could be used in lieu of a functional assessment. Anti-RBD antibodies are a major component of the neutralization response, as they noted, but antibodies that target other spike epitopes, such as S2, play a key role in other correlates of protection, such antibody-dependent cellular cytotoxicity.

There is no discussion about the C-reactive proteins and antibody levels. Is there any literature to support that or for comparison?

A more complete comparison of the antibody levels at diagnosis vs one year later would have strengthened the analysis, but it is understood that the sampling was not completed on all patients at Dx.

Specific comments:

Did the young age and severe disease overlap in this cohort? It would be good to understand if the age and severe disease associations with antibody levels are independent or related. Maybe there were high antibody levels in the younger patients because they were more sick?

Lines 89-90: Sentence needs to be updated. Assuming it should say anti-spike antibodies are more durable.

The spacing in Table 1 is off, specifically the last box with the blood parameter analysis.

Line 126: Might be more appropriate to say “serological” instead of “microbiological”

Line 265-6: “Patients with more severe symptoms had higher viral loads suggesting that the initial amounts of viral antigens may contribute to stronger serological responses.” If this is from data in the current study, those data should be reported.

Ethics statement and data availability statements are complete and adequate.

Reviewer 3 Report

Comments and Suggestions for Authors

General Comments:

This Spanish study examined SARS-CoV-2 antibody levels in youth under 18 years that had been infected with SARS-CoV-2 at least a year before. They focused on IgG levels in plasma. This likely underestimated the antibody response, since as a respiratory virus this would more likely evoke an IgM and IgA response, and saliva would be a better indicator of a mucosal response that would be more relevant with respect to immune protection. The study was focused on spike antibody responses, even though there are at least 27 other viral proteins encoded by the SARS-CoV-2 virus. Some limited analysis of anti-nucleocapsid antibody levels was performed. One advantage of studying antibody persistence in the children compared to adults with respect to SARS-CoV-2 antibodies is that they would be less likely to have cross-reactive antibodies from previous infections with coronaviruses. Pre-existing antibodies in adult serum from May – June, 2020 that can cross-react with SARS-CoV-2 spike and nucleocapsid proteins has been observed to be common [Majdoubi A, Michalski C, O'Connell SE, Dada S, Narpala S, Gelinas J, Mehta D, Cheung C, Winkler DF, Basappa M, Liu AC, Görges M, Barakauskas VE, Irvine M, Mehalko J, Esposito D, Sekirov I, Jassem AN, Goldfarb DM, Pelech S, Douek DC, McDermott AB, Lavoie PM. A majority of uninfected adults show preexisting antibody reactivity against SARS-CoV-2. JCI Insight. 2021 Apr 22;6(8):e146316. doi: 10.1172/jci.insight.146316. PMID: 33720905; PMCID: PMC8119195].

Specific Comments: 

1.    The immunogen of focus in this study was the Spike protein, and IgG antibodies against the full length recombinant Spike protein was used. The disadvantage of this is that the particular antibodies generated early on in the children may target different epitopes with repeated exposures as would have occurred a year later. From the way the assay was set up and performed, this would be impossible to tell.

2.    Since there are multiple waves of exposure to SARS-CoV-2 during the study period, persistence of IgG levels will be reflective of how many exposures occurred and when the last one was. Antibody levels normally decline when the immunogen is no longer in the environment, but IgG antibodies last at least 3-times longer than IgA and IgM antibodies.

3.    Lines 75 and 76 - The spike protein is not particularly immunogenic with respect to epitopes, but being a larger protein, it evokes a larger immune response. The nucleocapsid protein is poorly immunogenic, because about half of COVID-19-recovered patients had little detectable nucleocapsid antibodies early on in the pandemic. With repeated exposures, nucleocapsid antibodies became easier to detect serologically with Omicron infections. The most immunogenic SARS-CoV-2 structural protein appears to be the Membrane protein. This huge difference in the sensitivity between the anti-spike and anti-nucleocapsid antibody assays is reflected in the AU/ml cut-offs for detection described in lines 144-147.

4.    Lines 79-81 – IgG antibody levels in the nasopharyngeal area and upper lungs are relative low, and the main mucosal response, which is most protective, involves the production of IgM and IgA antibodies.

5.    Lines 89-91 – The statement about the durability of the nucleocapsid antibodies is repeated twice, but with a different reference in each case. The sensitivity of anti-nucleocapsid antibody assays are weaker than anti-spike-based assays, so with the decline in titre over time, this would be more problematic for detection. In any event, the nucleocapsid protein is located inside of the SARS-CoV-2 virus particle, and would be less important for antibody or T-cell recognition for taking out the virus. That is, the levels of anti-nucleocapsid antibodies are not reflective of direct protection, but of perhaps a generalized antibody response with other antibodies produced against the spike, membrane and envelope viral proteins, which are more confering important for immune protection.

6.    Lines 185-187 – “Regarding age, higher anti-spike IgG levels in those patients younger than 2 years (p=0.034) were observed when compared to the rest of the age groups.” How many of these children were being breast fed and may have received antibodies from their mothers?

7.    Lines 192-196 – IgG levels would be expected to be higher for detection with a lower lung infection than a SARS-CoV-2 infection confined to the nasopharynx and upper lungs.

8.    Line 212 – “None of the patients included were diagnosed of reinfection by SARS-CoV-2 during the follow-up.”  I think it would be more accurate to state “None of the patients included were diagnosed with COVID-19 during the follow-up.”  Unless the children were continually monitored with PCR or rapid antigen tests over the course of the year, it is impossible to know whether they had been reinfected by the virus. Some 45% of the pediatric patients were completely asymptomatic at the beginning of the study, but were apparently infected.

9.    Line 232 – That antibodies generated against SARS-CoV-1 or cold viruses prevent reinfections for only about a year is untrue. The cited reference #22 shows that antibody levels against SARS-CoV-1 lasted for at least a year. Again the authors are confusing getting re-infected with getting a disease.

10.    Line 324-326 – It seems that the authors copied the “Guidelines to Authors” rather than complete this section as it pertained to them.

11.     Reference List – There is unnecessary redundancy in the numbering of the references. There is inconsistency in the use of capitalization of the words used in the titles of the references from one reference to another. Sometimes each word is capitalized and in other cases, they are not.

Round 2

Reviewer 1 Report

Comments and Suggestions for Authors

Thank you for the revised version of the manuscript. All concerns have been addressed properly.

Author Response

We sincerely appreciate the reviewer’s comments. Their suggestions have significantly enhanced the quality and clarity of our manuscript.

Reviewer 2 Report

Comments and Suggestions for Authors

The authors have sufficiently addressed any concerns. They have added needed clarity and details to the methods and results.

Author Response

We are very thankful. We believe the revised version was much improved as a result of their valuable feedback.

Reviewer 3 Report

Comments and Suggestions for Authors

General Comments:

1.     The authors have addressed several of the concerns that I raised previously for their earlier version. However, there are still some minor issues that need to be fixed that are highlighted below.

Specific Comments:

1.     Page 1, line 48 – change “IgG anti-spike” to” anti-spike IgG”

2.     Page 2, lines 54 and 55 – change to “Higher anti-spike IgG levels were noted in those patients younger than 2 years..” 

3.     Page 2, lines 67-69. While the first MIS-C case may have been reported in 2020, it is likely that the disease occurred with previous coronavirus infections in children, but was rare and not identified as such.

4.     Page 2, line 76 – delete “On the other hand,” since it is not stated what was “on the one hand” and is superfluous. Same issue on page 8, lines 324 and 340.

5.     Page 2 - The amino acid sequence identity between SARS-CoV and SARS-CoV-2 nucleocapsid proteins is 90.5% [Zeng W, Liu G, Ma H, Zhao D, Yang Y, Liu M, Mohammed A, Zhao C, Yang Y, Xie J, Ding C, Ma X, Weng J, Gao Y, He H, Jin T. Biochemical characterization of SARS-CoV-2 nucleocapsid protein. Biochem Biophys Res Commun. 2020 Jun 30;527(3):618-623. doi: 10.1016/j.bbrc.2020.04.136. Epub 2020 Apr 30. Erratum in: Biochem Biophys Res Commun. 2022 Jul 23;614:225. doi: 10.1016/j.bbrc.2022.05.058] and for the spike protein is 81% [Pokhrel S, Kraemer BR, Burkholz S, Mochly-Rosen D. Natural variants in SARS-CoV-2 Spike protein pinpoint structural and functional hotspots with implications for prophylaxis and therapeutic strategies. Sci Rep. 2021 Jun 23;11(1):13120. doi: 10.1038/s41598-021-92641-x] Thus, the difference in conservation of the nucleocapsid verses spike protein between coronaviruses is not that dramatic, and the spike protein of SARS-CoV would be expected to elicit strong immune reponses also against the SARS-CoV-2 spike protein, which in fact is what we have observed in my own lab.

6.     Page 2, line 82-84 – The authors indicate that serum IgG is a good indicator of virus neutralization, whereas muscosal IgM and IgA are most effect in preventing viral entry. My understanding is that viral “neutralization” primarily refers to the blocking of spike protein to the ACE2 receptor and entry into hosts cell. Consequently, I don’t get the distinction that the authors seem to be inferring.

7.     Page 4, line 162 – Why is the higher threshold of 1.4 required for the nucleocapsid protein antibody detection than the 1.0 for the spike protein antibody?

8.     Page 4, line 165 – change “run” to “performed” to reduce jargon.

9.     Page 8, line 354 – change “conferred” to “obtained” or “experienced”

10.  Page 9, References – There is still inconsistency in the usage of capitalization in the titles of the cited articles (e.g., Ref. #8, 14, 17, 24, 26, 27, 30,

Comments on the Quality of English Language

The English is fine with the suggested corrections.
